# Genotype-Phenotype Correlation of Seven Known and Novel β-Globin Gene Variants

**DOI:** 10.3390/ijms26188872

**Published:** 2025-09-12

**Authors:** Kritsada Singha, Anupong Pansuwan, Goonnapa Fucharoen, Supan Fucharoen

**Affiliations:** 1Centre for Research and Development of Medical Diagnostic Laboratories, Faculty of Associated Medical Sciences, Khon Kaen University, Khon Kaen 40002, Thailand; kritsada.si@msu.ac.th (K.S.); anuppa@kku.ac.th (A.P.); goonnapa@kku.ac.th (G.F.); 2Biomedical Science Research Unit, Faculty of Medicine, Mahasarakham University, Mahasarakham 44000, Thailand

**Keywords:** beta globin gene, single-nucleotide polymorphisms, phenotype-genotype interactions

## Abstract

Variants of uncertain significance (VUS) are often challenging for genetic counseling and require additional data for accurate variant classification. This study aims to describe the genotype-phenotype correlation of the seven β-globin gene variants found in Thailand. Retrospective data in a total of 45,914 subjects encountered at our diagnostic laboratory from January 2012 to December 2024 were reviewed. A total of 33 leftover EDTA blood specimens, suspected of having β-globin gene defects, were included. Eighty-nine normal subjects were also analyzed to confirm phenotypic expression of the variants. The whole β-globin and Krüppel-like factor 1 (*KLF1*) genes were examined using PCR-based methods. Seven nucleotide variants were identified among 33 suspected subjects, including a novel (β^−206(C>G)^), four hitherto undescribed in Thailand [β^−198(A>G)^, β^IVSII−180(T>C)^, β^IVSII−337(A>G)^, and β^*233(G>C)^], and two known variants [β^−50(G>A)^ and β^IVSII−258(G>A)^]. The β^−198(A>G)^ and β^*233(G>C)^ variants were also identified in 1.69% of normal subjects, indicating neutral DNA polymorphisms. All subjects of β^−198(A>G)^, β^IVSII−180(T>C)^, β^IVSII−258(G>A)^, and β^IVSII−337(A>G)^ with borderline Hb A_2_ levels had *KLF1* mutations. Compound heterozygous β^−206(C>G)^ and known β^+^-thalassemia trait revealed β-thalassemia trait phenotype. In silico pathogenicity prediction showed that the β^−206(C>G)^, β^−198(A>G)^, β^IVSII−180(T>C)^, β^IVSII−258(G>A)^, β^IVSII−337(A>G)^, and β^*233(G>C)^ were associated with benign variants. It was found that heterozygous β^−50(G>A)^ had elevated Hb A_2_ levels resembling those of β-thalassemia trait. However, the association of the β^−50(G>A)^ and Hb E or β-thalassemia revealed a phenotype of Hb E or β-thalassemia trait. Most prediction tools indicate that the β^−50(G>A)^ is associated with benign variants; however, PromoterAI revealed that the β^−50(G>A)^ is associated with under-expression of the β-globin gene with high sensitivity. Based on these findings, the β^−50(G>A)^ is most likely a very mild β^+^-thalassemia allele. This study described the genotype-phenotype correlation of known and novel β-globin gene variants found in Thailand. The data should prove useful for accurate variant classification, genetic counseling, and a prevention and control program of severe thalassemia diseases in Thailand.

## 1. Introduction

β-Thalassemia is a group of genetic diseases characterized by defects of β-globin chain production. Two main types, β^0^- and β^+^-thalassemia, are, respectively, defined based on the absence or reduction in β-globin chain synthesis. While no β-globin chain production is β^0^-thalasemia, a reduction in β-globin chain synthesis is β^+^-thalassemia. A very mild β^+^-thalassemia allele is sometimes referred to as β^++^-thalassemia [1,2]. This depends mainly on the β-globin gene defect, which has implications for genetic counseling and prenatal diagnosis of the disease. Heterozygous β-thalassemia is associated with raised Hb A_2_ level and reduced mean corpuscular volume (MCV) and mean corpuscular hemoglobin (MCH), whereas the homozygote and compound heterozygote with other β-hemoglobinopathies can express as various clinical symptoms, ranging from mild to severe hemolytic anemia, jaundice, hepatosplenomegaly, and iron overload [1]. In Thailand, the average frequency of α-thalassemia is 20–30%, β-thalassemia is 3–9%, and that of Hb E is 30–50%. The interaction of these genes could lead to several thalassemia syndromes with variable phenotypic expression ranging from mild thalassemia intermedia to severe transfusion-dependent thalassemia [3].

Normal human hemoglobin (Hb) is a heterotetramer that contains two pairs of different globin chains, including the α-like (ζ- or α-chains) and β-like (ε-, γ-, δ-, or β-chains) globin chains. Adult Hb (Hb A), consisting of two α- and two β-globin chains (α_2_β_2_), is the major human Hb, followed by 2–3% of Hb A_2_ (α_2_δ_2_) and less than 1% of Hb F (α_2_γ_2_), after one year of birth. Many nucleotide variants of globin genes causing thalassemia and hemoglobinopathies have been described worldwide [1]. Most known thalassemia and hemoglobinopathies defects have been found in the β-globin and the α-globin genes due to the major adult Hb components [2]. Although most of them are associated with known phenotypic expressions, some variants lack sufficient information to support a more definitive classification and are therefore classified as variants of uncertain significance (VUS). VUS is often challenging for genetic counseling and requires additional data for accurate variant classification [4,5]. At our routine thalassemia diagnostics, many subjects were identified as having known and novel VUS in the β-globin gene, leading to inconsistencies between genotype and phenotype and a difficulty in providing genetic counseling. This study aims to describe the genotype-phenotype correlation of the seven β-globin gene variants, including β^−206(C>G)^, β^−198(A>G)^, β^−50(G>A)^, β^IVSII−180(T>C)^, β^IVSII−258(G>A)^, β^IVSII−337(A>G)^, and β^*233(G>C)^, found in Thailand.

## 2. Results

Seven β-globin gene variants were identified during routine thalassemia investigation of 33 referral subjects to our laboratory, including a novel one [β^−206(C>G)^ (HBB:c.-256C>G, rs376005360, IthaID: 4153)], four hitherto undescribed in Thailand [β^−198(A>G)^, β^IVSII−180(T>C)^, β^IVSII−337(A>G)^, and β^*233(G>C)^], and two known variants [β^−50(G>A)^ and β^IVSII−258(G>A)^] [2,6,7,8,9,10,11,12]. The β^−198(A>G)^, β^IVSII−180(T>C)^, β^IVSII−337(A>G)^, and β^*233(G>C)^ have been described in various ethnic groups, such as Azerbaijani, Chinese, Malay, Ibany, and Palestinian [2,6,7,8,9,10,11,12]. The corresponding DNA sequencing profiles are shown in Figure 1. Table 1 lists data of the subjects, including Hb and hematological parameters, α- and β-globin genotypes, and *KLF1* mutations. The first three subjects (Table 1, no. 1–3) carried novel variant β^−206(C>G)^ in different combinations. All of them had elevated Hb A_2_ levels (4.4–5.1%). β-Globin gene analysis identified a compound heterozygote state in all of them, i.e., [β^−206(C>G)^/β^−31(A>G)^] in subjects no. 1 and 2 and [β^−206(C>G)^/ β^−50(G>A)^] in subject no. 3. Allele-specific amplification of the β^−206(C>G)^ followed by DNA sequencing confirmed that the β^−206(C>G)^ and β-thalassemia mutations are located *in trans* (Figure 2). The representative DNA sequencing profiles are shown in Appendix A. α-Globin genotyping revealed that subject no. 1 also carried a heterozygous Hb Constant Spring (α^CS^α) gene. As compared to those of subjects with corresponding mutations in our series, we did not observe that co-inheritance of this β^−206(C>G)^ with the β^−31(A>G)^ β^+^-thalassemia has worsened the hematological phenotype as compared to the patients with β^−31(A>G)^ β^+^-thalassemia alone.

Six subjects (no. 4–9) were found to be heterozygous for β^−198(A>G)^, involving three subjects with normal Hb A_2_ levels (2.8–2.9%), and another three subjects with borderline Hb A_2_ levels (3.6–3.9%) (Table 2). Most of them had relatively normal hematological parameters, although three of them (no. 4, 7, and 9) also carried α^+^-thalassemia (3.7 kb deletion). Notably, it was found that three of them with borderline Hb A_2_ levels (no. 7–9) were carriers of three different *KLF1* mutations (NC_000019.10), c.892G>C (A298P), c.525_526insGCGCCGG (G176AfsX179), and c.895C>G (H299D), previously described in Thailand [13,14].

As shown in Table 1, β^−50(G>A)^ is the most common variant found in this study. It was detected as a heterozygous state in 9 subjects (no. 10–18), a compound heterozygote with β-thalassemia in 4 subjects (no. 19–22), and in compound heterozygosis with β-thalassemia and Hb E mutations in 5 subjects (no. 23–27). The β^−50(G>A)^ and Hb E mutations are located *in trans*, as confirmed by allele-specific amplification and DNA sequencing (Figure 2). Most of the subjects with heterozygous β^−50(G>A)^ had borderline Hb A_2_ levels (3.6–4.1%). An exception was noted in one case (no.10) with Hb A_2_ 2.0% who was a carrier of α^+^-thalassemia (3.7 kb deletion). In this group of subjects, two of them (no. 15 and 16) also carried heterozygous *KLF1* mutations, i.e., c.525_526insGCGCCGG and c.983G>A (R328H). It is expected that the four subjects with compound β^−50(G>A)^/β-thalassemia (no. 19–22) had a β-thalassemia trait phenotype with slightly decreased mean corpuscular volume (MCV) and mean corpuscular hemoglobin (MCH) levels. All (no. 24–27) but one case (no. 23) of compound β^−50(G>A)^/β^E^ exhibited the phenotype of Hb E trait. This case, no. 23 with more severe anemia (Hb 8.0 g/dL), was in fact the patient with Hb H-ConSp disease in combination with the β^−50(G>A)^/β^E^ genotype, known as the ConSpEABart’s disease, commonly encountered in Thai populations [15,16]. As for the β^−206(C>G)^, we found that co-inheritance of this β^−50(G>A)^ with β-thalassemia or Hb E has not worsened the hematological phenotype as compared to the patients with β-thalassemia or Hb E alone. In addition, when the β^−50(G>A)^/β^E^ genotype (n = 4) was compared to the β^−28(G>A)^/β^E^ in our series (n = 143), it was apparent that the latter had a more severe hematological phenotype, i.e., lower Hb, MCV, MCH levels, and higher red cell distribution width (RDW), Hb E, and Hb F levels significantly. The β^−50(G>A)^/β^E^ subjects (n = 4) had no significant difference in hematological parameters from those of subjects with Hb E traits in our series (β^E^/β^A^) (n = 112), except for the higher Hb F level (Table 2). Some insignificant statistical differences in the two groups may be due to the small sample size.

Subjects no. 28–30 represented cases with rarer β-variants in this study, including β^IVSII−180(T>C)^, β^IVSII−258(G>A)^, and β^IVSII−337(A>G)^. Although hematological parameters are not completely available, they all had borderline Hb A_2_ levels (3.6–3.8%). Further *KLF1* gene analysis identified that all of them were carriers of *KLF1* mutations, the c.892G>C (n = 2) and c.983G>A (n = 1). Molecular analysis of the last group (case no. 31–33), who had normal hematological features and Hb A_2_ levels (2.6–2.8%), identified that all of them were carriers of the β^*233(G>C)^ variant [233 bp after stop codon of β-globin gene or TTS+99 (G>C)]. Only case no. 31 was found additionally to be a carrier of α^+^-thalassemia (3.7 kb deletion). In this study, we did not detect α-globin gene triplication (ααα^anti−3.7^) in any of the subjects. Among 89 normal control subjects, three were found to be carriers of the β^−198(A>G)^ and the β^*233(G>C)^ variants, the data indicating neutral DNA polymorphisms of these two SNPs.

Table 3 summarizes allele frequencies of neutral and seven β-globin gene variants identified in this study, as compared globally using reference SNP (rs) reported on the National Center for Biotechnology Information (NCBI) (https://www.ncbi.nlm.nih.gov/). As shown in the table, high allele frequencies include β^CD2(CAT>CAC)^, β^IVSII−16(G>C)^, β^IVSII−74(T>G)^, β^IVSII−81(C>T)^, β^IVSII−666(C>T)^, and β^*233(G>C)^, whereas β^−206(C>G)^, β^−198(A>G)^, β^−50(G>A)^, β^IVSII−180(T>C)^, β^IVSII−258(G>A)^, and β^IVSII−337(A>G)^ represent those with low allele frequencies. In our study on normal Thai population, β^−198(A>G)^, β^CD2(CAT>CAC)^, β^IVS II−16(G>C)^, β^IVSII−74(T>G)^, β^IVSII−81(C>T)^, β^IVSII−666(C>T)^, and β^*233(G>C)^ were detected at the allele frequencies of more than 1%, whereas those of β^−206(C>G)^, β^−50(G>A)^, β^IVSII−180(T>C)^, β^IVSII−258(G>A)^, and β^IVSII−337(A>G)^ were not observed. It is most likely that these variants have very low frequencies in general populations. Appendix A reveals no association between Hb A_2_ levels and variants found in the Thai population.

Table 4 summarizes the results of in silico functional prediction for the seven β-globin gene variants identified using SpliceAI, CADD, PhyloP, and PromoterAI. These analyses predict that β^−206(C>G)^, β^−198(A>G)^, β^IVSII−180(T>C)^, β^IVSII−258(G>A)^, β^IVSII−337(A>G)^, and β^*233(G>C)^ are associated with benign variants. The β^−50(G>A)^ is also associated with benign variants by the SpliceAI, CADD, and PhyloP programs, but is found to be under-expressed with high sensitivity using the PromoterAI program. For the online TFBIND, many transcription factor binding sites were found to be altered in the promoter β-globin variants, particularly the CCAAT box in β^−206(C>G)^, GATA-1 in β^−198(A>G)^, and Sp1, GATA-1, and CP1 in β^−50(G>A)^ (Appendix A). For splice site prediction, no difference was observed using the Neural Network in NNSPLICE 0.9 version and FGENESH for the β^IVSII−180(T>C)^, β^IVSII−258(G>A)^, β^IVSII−337(A>G)^, and wild-type sequences.

## 3. Discussion

For globin genes, numerous single-nucleotide polymorphisms (SNPs) have been identified, particularly in the β-globin gene. The β^CD2(CAT>CAC)^, β^IVSII−16(G>C)^, β^IVSII−74(T>G)^, β^IVSII−81(C>T)^, and β^IVSII−666 (C>T)^ are the common SNPs within the β-globin gene. During our routine thalassemia investigation, a novel and six known variants in the β-globin gene, including β^−206(C>G)^, β^−198(A>G)^, β^−50(G>A)^, β^IVSII−180(T>C)^, β^IVSII−258(G>A)^, β^IVSII−337(A>G)^, and β^*233(G>C)^, were identified with genotype-phenotype inconsistency and need to be proven for appropriate variant classification. They are distributed and located in the promoter, intervening sequence, and 3ʹ end of the β-globin gene. Of these, the β^−206(C>G)^ is a novel one. This novel β^−206(C>G)^ and five known variants, including β^−198(A>G)^, β^IVSII−180(T>C)^, β^IVSII−258(G>A)^, β^IVSII−337(A>G)^, and β^*233(G>C)^, should be considered as benign or likely benign variants due to the following reasons. First, the combination of the β^−206(C>G)^ *in trans* to known β^+^-thalassemia mutations results in a β-thalassemia trait phenotype, which is distinct from the homozygous β-thalassemia phenotype, as compared in Table 2. Second, allele frequencies of the β^−198(A>G)^ and the β^*233(G>C)^ were more than 1% in normal control subjects, whereas no β^−206(C>G)^, β^−50(G>A)^, β^IVSII−180(T>C)^, β^IVSII−258(G>A)^, and β^IVSII−337(A>G)^ were observed. It is most likely that the variants have very low frequencies in the general population. Third, all subjects with β^−198(A>G)^, β^IVSII−180(T>C)^, β^IVSII−258(G>A)^, and β^IVSII−337(A>G)^, who had borderline Hb A_2_ levels, carried *KLF1* mutations known to be associated with increased Hb A_2_ levels [13,14]. Fourth, no difference in human splicing sites was predicted among the wild-type, β^IVSII−180(T>C)^, β^IVSII−258(G>A)^, and β^IVSII−337(A>G)^. Fifth, previous studies have identified that β^−198(A>G)^, β^IVSII−180(T>C)^, β^IVSII−258(G>A)^, β^IVSII−337(A>G)^, and β^*233(G>C)^ are associated with benign phenotypes, characterized by normal MCV, MCH, and Hb A_2_ levels in the heterozygotes [6,7,9,10,11,12]. Lastly, all functional prediction programs indicated that these six variants are benign or likely benign variants. However, according to the nucleotide changes, it is conceivable that the β^−206(C>G)^ and β^−198(A>G)^ may alter erythroid-specific transcription factor binding sites, especially the CCAAT box (CCAAG to CCAAC) for β^−206(C>G)^ and homologous GATA-1 binding motif (AGATAT to AGATGT) for β^−198(A>G)^, respectively (Appendix A).

This contrasts with the β^−50(G>A)^, which was first reported in Chinese and later in Thai and Malaysian patients [8,11,17,18,19]. It is also the most common β-globin variant found in this study, being detected in 18 of 33 subjects (Table 1). This β^−50(G>A)^ variant is located within the highly conserved direct repeat element (DRE), AGGGCAGGAGCCAGGGCTGGGC, between the CCAAT and TATA elements in the human β-globin promoter. It has been documented that heterozygous β^−50(G>A)^ had microcytic hypochromic RBCs with increased Hb A_2_ (4.6%) and Hb F (1.2%) levels, and compound heterozygous of β^−50(G>A)^ and β^0^-thalassemia revealed moderate microcytic hypochromic anemia with high Hb F levels (27.7%), and a β^0^/β^+^ thalassemia phenotype. This finding suggests that the β^−50(G>A)^ is associated with a β^+^-thalassemia [8]. However, there is a study showing that most cases of heterozygous β^−50(G>A)^ are associated with normochromic normocytic RBCs and normal Hb A_2_ levels [17]. Another study reported that, as compared to the wild-type, heterozygous β^−50(G>A)^ had decreased MCV and MCH values and slightly elevated Hb A_2_ levels [18]. Moreover, compound heterozygous β^−50(G>A)^ and β^0^-thalassemia revealed a β-thalassemia trait phenotype [18]. Our study favors the latter. As shown in Table 1, most heterozygous β^−50(G>A)^ revealed microcytic hypochromic RBCs with borderline Hb A_2_ levels (3.6–4.1%), no matter with or without *KLF1* mutation. However, in the β^−50(G>A)^/β^0^ and β^−50(G>A)^/β^+^ genotypes, all patients revealed a β-thalassemia trait phenotype, rather than a severe homozygous β-thalassemia phenotype. Similarly, our subjects with β^−50(G>A)^/β^E^ had a very similar phenotype to that of the Hb E trait, rather than a β^+^/β^E^ thalassemia phenotype (Table 2). It is noteworthy that no β^−50(G>A)^ was identified among normal control subjects. In addition, most prediction tools indicate that the β^−50(G>A)^ is associated with benign variants; however, PromoterAI revealed that the β^−50(G>A)^ is associated with under-expression with high sensitivity. Moreover, altered transcription factor binding sites, particularly those of Sp1, GATA-1, and CP1, were identified. Accordingly, based on these data, our data support that the β^−50(G>A)^ should be classified as a very mild β^+^-thalassemia allele, corresponding to the β^++^ or β^+++^ phenotype.

In conclusion, we have identified a novel and six known β-globin gene variants in Thai individuals. Based on genotype-phenotype correlation, in silico functional analysis, frequency of each variant in normal control group, as well as accumulated data from previous studies, it is concluded that six of them including β^−206(C>G)^, β^−198(A>G)^, β^IVSII−180(T>C)^, β^IVSII−258(G>A)^, β^IVSII−337(A>G)^, and β^*233(G>C)^ are benign β-globin gene variants or normal phenotype (non-β-thalassemia), whereas β^−50(G>A)^ is a β^++^ or β^+++^ thalassemia allele. These data should prove useful for accurate variant pathogenicity classification, genetic counselling, and the prevention and control program of severe thalassemia diseases in the region.

## 4. Materials and Methods

### 4.1. Subjects and Hematological Analyses

Ethical approval of the study protocol on human research was obtained by the Institutional Review Board (IRB) of Khon Kaen University, Thailand (HE682074). Retrospective data in a total of 45,914 subjects encountered at our center for thalassemia diagnosis from January 2012 to December 2024 were reviewed. A total of 33 leftover DNA specimens with inconsistencies between genotype and phenotype were selectively recruited. Eighty-nine normal control subjects (178 alleles) with Hb type A_2_A, normal Hb A_2_ levels (2.2–3.2%), and normal hematological parameters (non-anemia, MCV of 80–100 fL, and MCH of 27–33 pg) were also analyzed to confirm phenotypic expression of the variants. The β-thalassemia allele should not be identified in this group of normal control subjects. Hematological parameters were recorded on standard blood cell counters. Hb analysis was carried out by using capillary electrophoresis (Capillarys II Flex Piercing, Sebia, France) or high-performance liquid chromatography (HPLC) (Variant^TM^, Bio-Rad Laboratory, Hercules, CA, USA).

### 4.2. Molecular Analysis

Common β-thalassemia genes, α-thalassemia mutations (--^SEA^, --^THAI^, −α^3.7^, −α^4.2^, Hb Constant Spring [HBA2:c.427T>C], and Hb Paksé [HBA2:c.429A>T]), α-globin gene triplication (ααα^anti−3.7^), and *KLF1* mutations previously described in Thailand were characterized by PCR-based methods as described previously [3,13,14,20,21]. DNA sequencing of the whole β-globin gene was done using an ABI PRISM^TM^ 3730 XL analyzer (Applied Biosystems, Foster City, CA, USA) and Barcode-tagged sequencing (BTSeq^TM^, Celemics, Korea).

### 4.3. Molecular Confirmation of the β^−206(C>G))^ and β^−50(G>A)^

To confirm the configurations (*in cis* or *in trans* location) of the β^−206(C>G)^ and the β^−50(G>A)^, found in the subject, allele-specific amplification was performed. The forward primers G220 (5′-TGTACTGATGGTATGGGGCG-3′) for β^−206(C>G)^ specific allele and G118 (5′-ACTCCCAGGAGCAGGGAGGA-3′) for β^−50(G>A)^ specific allele were amplified separately in two allele-specific PCR conditions with the common reverse primer S3 (5′-TCCCATAGACTCACCCTGAA-3′) to produce specific fragments of 734 bp and 580 bp, respectively. Normal subjects (β^A^/β^A^) were used as a negative control. Allele-specific DNA sequencing of the amplified fragment was then carried out.

### 4.4. Transcription Factor Binding Site Prediction for β^−206(C>G)^, β^−198(A>G)^, and β^−50(G>A)^

Transcription factor binding site prediction for β^−206(C>G)^, β^−198(A>G)^, and β^−50(G>A)^ was examined by using online TFBIND (https://tfbind.hgc.jp/, accessed 24 April 2025) [22]. The similarity score related to the transcription factor binding affinity of the variant sequences was compared to the wild-type sequences (Appendix A).

### 4.5. Splice Site Prediction of β^IVSII−180(T>C)^, β^IVSII−258(G>A)^, and β^IVSII−337(A>G)^

Human splice sites and the potential genes and exons in the human genomic DNA of β^IVSII−180(T>C)^, β^IVSII−258(G>A)^, and β^IVSII−337(A>G)^ as compared to wild-type were predicted by Neural Network in NNSPLICE 0.9 version (https://www.fruitfly.org/seq_tools/splice.html, accessed 24 April 2025) [23] and the FGENESH 2.6 program (http://www.softberry.com/berry.phtml, accessed 24 April 2025) [24].

### 4.6. In Silico Functional Study Prediction for the Seven β-Globin Variants

In silico functional study prediction for pathogenicity classification of the seven β-globin variants found was analyzed by SpliceAI, Combined Annotations Dependent Depletion (CADD), PhyloP, and PromoterAI (https://spliceailookup.broadinstitute.org/, accessed 24 April 2025) [25,26,27].

### 4.7. Statistical Analysis

Hematological data were analyzed using the STATA^TM^ version 18 (Stata Corp, College Station, TX, USA). Non-parametric statistic was selected due to the non-normal distribution of data. The difference between the two independent groups was compared by the Mann–Whitney U test, with statistical significance defined as *p*-value < 0.05.

## Figures and Tables

**Figure 1 ijms-26-08872-f001:**
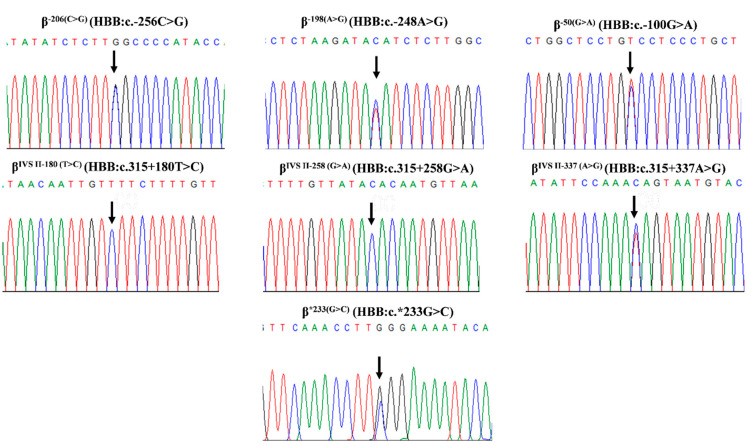
DNA sequencing profiles of a novel and six known β-globin variants, demonstrating β^−206(C>G)^, β^−198(A>G)^, β^−50(G>A)^, β^IVSII−180(T>C)^, β^IVSII−258(G>A)^, β^IVSII−337(A>G)^, and β^*233(G>C)^. Abbreviation: *HBB*, hemoglobin subunit beta.

**Figure 2 ijms-26-08872-f002:**
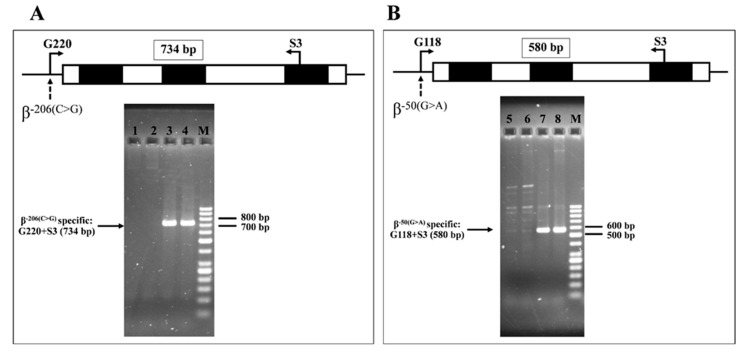
Allele-specific amplification of β^−206(C>G)^ (**A**) and β^−50(G>A)^ (**B**). The specific β^−206(C>G)^ allele was generated using primers G220 and S3 (**A**), and the β^−50(G>A)^ allele was generated using primers S118 and S3 (**B**), to produce specific fragments of 734 bp and 580 bp, respectively. Normal subjects (β^A^/β^A^) were used as a negative control for the two conditions. M represents the GeneRuler^TM^ 50 bp DNA ladder. Lanes 1, 2, 5, and 6: normal control subjects (β^A^/β^A^) in the two conditions, lanes 3 and 4: β^−206(C>G)^ positive subjects, and lanes 7 and 8: β^−50(G>A)^ positive subjects.

**Table 1 ijms-26-08872-t001:** Hb analysis, hematological parameters, and molecular analysis of subjects with seven β-globin gene variants.

No	Sex ^a^	Hb Type	Hb A_2_ (%)	Hb E + A_2_ (%)	Hb F (%)	RBC (10^12^/L)	Hb (g/dL)	Hct (%)	MCV (fL)	MCH (pg)	MCHC (g/dL)	RDW (%)	β-Globin Genotype	α-Globin Genotype	*KLF1* Mutation Identified ^b^
1	M	A_2_A	4.4	-	0.6	6.0	16.2	46.2	77.4	27.1	35.1	14.8	β^−206(C>G)^/β^−31(A>G)^	α^CS^α/αα	-
2	M	A_2_A	4.5	-	0	6.1	13.3	39.1	64.5	21.9	34.0	18.3	β^−206(C>G)^/β^−31(A>G)^	αα/αα	-
3	F	A_2_A	5.1	-	0.7	5.3	10.9	34.7	65.2	20.4	31.3	18.6	β^−206(C>G)^/β^−50(G>A)^	αα/αα	-
4	F	A_2_A	2.9	-	0	4.7	12.9	38.7	83.0	27.5	33.2	12.1	β^−198(A>G)^/β^A^	−α^3.7^/αα	-
5	M	A_2_A	2.9	-	0	4.9	14.2	42.1	86.8	29.2	33.6	13.2	β^−198(A>G)^/β^A^	αα/αα	-
6	M	A_2_A	2.8	-	0	4.8	15.1	43.8	90.8	31.4	34.6	12.4	β^−198(A>G)^/β^A^	αα/αα	-
7	M	A_2_A	3.6	-	0	Na	Na	Na	78.4	25.5	32.5	Na	β^−198(A>G)^/β^A^	−α^3.7^/αα	A298P
8	M	A_2_A	3.6	-	2.2	Na	Na	Na	86.6	28.8	33.3	Na	β^−198(A>G)^/β^A^	αα/αα	G176AfsX179
9	F	A_2_A	3.9	-	2.4	5.8	13.6	42.2	72.8	23.4	32.2	Na	β^−198(A>G)^/β^A^	−α^3.7^/αα	H299D
10	F	A_2_(F)A	2.0	-	8.6	4.7	11.2	35.5	75.1	23.7	31.6	17.3	β^−50(G>A)^/β^A^	−α^3.7^/αα	-
11	M	A_2_A	3.6	-	0.3	Na	Na	Na	Na	Na	Na	Na	β^−50(G>A)^/β^A^	Na	-
12	F	A_2_A	3.6	-	1.5	Na	Na	Na	70.3	22.7	32.3	Na	β^−50(G>A)^/β^A^	αα/αα	-
13	F	A_2_A	3.7	-	0.6	Na	Na	Na	80.7	26.9	33.3	Na	β^−50(G>A)^/β^A^	−α^3.7^/αα	-
14	M	A_2_A	3.8	-	0.3	5.2	14.0	41.0	79.0	27.0	34.2	14.4	β^−50(G>A)^/β^A^	αα/αα	-
15	F	A_2_A	3.9	-	1.1	4.8	12.1	36.0	75.0	25.2	33.6	Na	β^−50(G>A)^/β^A^	−α^3.7^/αα	G176AfsX179
16	F	A_2_A	3.9	-	5.2	5.4	12.5	40.0	73.7	23.1	31.3	15.1	β^−50(G>A)^/β^A^	−α^3.7^/αα	R328H
17	F	A_2_A	4.1	-	0	Na	Na	Na	74.0	23.0	31.1	Na	β^−50(G>A)^/β^A^	Na	-
18	M	A_2_A	4.1	-	2.3	5.5	14.3	42.6	77.5	26.0	33.6	Na	β^−50(G>A)^/β^A^	Na	-
19	F	A_2_A	5.0	-	1.2	5.5	12.0	37.0	66.2	21.7	32.8	15.4	β^−50(G>A)^/β^Malay^	αα/αα	-
20	F	A_2_A	5.5	-	0	5.2	11.2	34.0	65.1	21.6	33.2	14.1	β^−50(G>A)^/β^41/42(-TTCT)^	αα/αα	-
21	F	A_2_A	6.1	-	1.7	4.4	8.3	26.9	61.0	18.9	31.0	17.6	β^−50(G>A)^/β^IVSI−1(G>T)^	αα/αα	-
22	M	A_2_A	6.2	-	0.6	5.9	12.9	40.0	67.4	20.6	32.4	Na	β^−50(G>A)^/β^IVSI−1(G>T)^	−α^3.7^/αα	-
23	M	CSEABart’s	2.3	16.3	1.7	5.1	8.0	29.2	57.7	15.8	27.4	23.5	β^−50(G>A)^/β^E^	--^SEA^/α^CS^α	-
24	M	E(F)A	-	27.7	6.8	6.2	14.2	44.7	72.0	22.7	31.5	18.2	β^−50(G>A)^/β^E^	−α^4.2^/αα	-
25	F	EA	-	28.2	3.3	5.0	13.3	40.0	79.5	26.5	33.3	15.0	β^−50(G>A)^/β^E^	αα/αα	-
26	F	E(F)A	-	29.7	6.2	Na	Na	Na	Na	Na	Na	Na	β^−50(G>A)^/β^E^	αα/αα	-
27	M	EA	3.4	31.5	1.1	Na	Na	Na	75.0	Na	Na	Na	β^−50(G>A)^/β^E^	Na	-
28	F	A_2_A	3.6	-	0	Na	Na	Na	74.9	24.5	32.7	Na	β^IVSII−180(T>C)^/β^A^	αα/αα	A298P
29	F	A_2_A	3.8	-	0	Na	Na	Na	Na	Na	Na	Na	β^IVSII−258(G>A)^/β^A^	αα/αα	A298P
30	F	A_2_A	3.6	-	5.0	Na	Na	Na	78.5	Na	Na	Na	β^IVSII−337(A>G)^/β^A^	−α^3.7^/αα	R328H
31	F	A_2_A	2.8	-	0	4.0	12.3	38.1	94.3	30.6	32.4	13.7	β^*233(G>C)^/β^A^	−α^3.7^/αα	-
32	M	A_2_A	2.6	-	0	5.4	15.6	45.0	83.3	28.9	34.7	12.8	β^*233(G>C)^/β^A^	αα/αα	-
33	F	A_2_A	2.7	-	0	4.2	12.4	36.3	86.3	29.4	34.1	13.3	β^*233(G>C)^/β^A^	αα/αα	-

^a^ All subjects are adults, except subject no. 26, who is 1.5 years old. ^b^ *KLF1* mutation found in a heterozygous state. Na: not available due to the lack of the indicated hematological data. Abbreviation: RBC, red blood cell; Hct, hematocrit; MCHC, mean corpuscular hemoglobin concentration; RDW, red cell distribution width.

**Table 2 ijms-26-08872-t002:** Hematological parameters and Hb analysis of β^−206(C>G)^ and β^−50(G>A)^ in heterozygotes and their combinations compared to other β-hemoglobinopathies.

β-Globin Genotype	n	Hb A_2_ (%)	Hb E + A_2_ (%)	Hb F (%)	RBC (10^12^/L)	Hb (g/dL)	MCV (fL)	MCH (pg)	RDW (%)
β^−206(C>G)^/β^−31(A>G)^	2	4.4, 4.5	-	0.6, 0	6.0, 6.1 ^a^	16.2, 13.3	77.4, 64.5	27.1, 21.9	14.8, 18.3
β^A^/β^−31(A>G)^	41	4.7 ± 0.4	-	1.2 ± 0.1	5.0 ± 0.7 ^a^	12.9 ± 1.5	77.4 ± 4.5	25.4 ± 1.3	14.7 ± 1.2
β^−50(G>A)^/β^A^	9	3.6 ± 0.3 ^b^	-	2.2 ± 2.9 ^b^	5.1 ± 0.4	12.8 ± 1.3 ^b^	75.7 ± 3.3 ^b^	24.7 ± 1.8 ^b^	15.6 ± 1.5 ^b^
β^A^/β^A^	89	2.8 ± 0.2 ^b^	-	0.1 ± 0.2 ^b^	5.0 ± 0.5	14.8 ± 1.6 ^b^	88.8 ± 3.9 ^b^	29.8 ± 1.4 ^b^	12.9 ± 0.9 ^b^
β^−50(G>A)^/β^Malay^	1	5.0	-	1.2	5.53	12.0	66.2	21.7	15.4
β^A^/β^Malay^	29	4.4 ± 0.4	-	1.3 ± 1.4	5.2 ± 0.8	12.0 ± 1.6	70.2 ± 5.1	23.4 ± 2.5	15.6 ± 1.8
β^−50(G>A)^/β^0^	3	5.9 ± 0.4	-	0.8 ± 0.9	5.2 ± 0.8	10.8 ± 2.3	64.5 ± 3.2	20.4 ± 1.4	15.9 ± 2.5
β^A^/β^0^	309	5.7 ± 0.7	-	1.4 ± 1.1	5.5 ± 0.9	11.4 ± 1.9	63.6 ± 4.0	20.6 ± 1.8	17.2 ± 2.0
β^−50(G>A)^/β^E^	4	-	29.3 ± 1.7 ^d^	4.4 ± 2.7 ^c,d^	5.6 ± 0.8	13.8 ± 0.6 ^d^	75.5 ± 3.8 ^d^	24.6 ± 2.7 ^d^	16.6 ± 2.3 ^d^
β^A^/β^E^	112	-	29.1 ± 3.0	1.4 ± 1.3 ^c^	5.1 ± 0.9	13.0 ± 1.7	77.3 ± 5.2	25.8 ± 2.2	14.7 ± 1.2
β^−28(G>A)^/β^E^	143	-	56.2 ± 6.7 ^d^	15.0 ± 8.2 ^d^	5.1 ± 0.9	9.7 ± 1.4 ^d^	60.8 ± 5.8 ^d^	19.2 ± 1.7 ^d^	21.9 ± 3.3 ^d^

^a^ Significant difference between β^−206(C>G)^/β^−31(A>G)^ and β^A^/β^−31(A>G)^. ^b^ Significant difference between β^−50(G>A)^/β^A^ and β^A^/β^A^. ^c^ Significant difference between β^−50(G>A)^/β^E^ and β^A^/β^E^. ^d^ Significant difference between β^−50(G>A)^/β^E^ and β^−28(G>A)^/β^E^.

**Table 3 ijms-26-08872-t003:** β-globin variants in Thai normal control and global subjects.

Variants	HGVS Name (*HBB*)	RS Number	Normal Thai Subjects			Global (gnomAD v4-Genomes)		
			N	Reference Allele	Alternative Allele	N	Reference Allele	Alternative Allele
−206(C>G)	c.−256C>G	rs376005360	178	1.00 (n = 178)	0.00 (n = 0)	149,210	0.999906	0.000094
−198(A>G)	c.−248A>G	rs76306358	178	0.9831 (n = 175)	0.0169 (n = 3)	149,276	0.999571	0.000429
−50(G>A)	c.−100G>A	rs281864524	178	1.00 (n = 178)	0.00 (n = 0)	149,276	0.999967	0.000033
CD 2 (CAT>CAC)	c.9T>C	rs713040	178	0.4213 (n = 75)	0.5787 (n = 103)	149,134	0.184794	0.815206
IVS II-16 (G>C)	c.315 + 16G>C	rs10768683	178	0.4213 (n = 75)	0.5787 (n = 103)	149,086	0.182519	0.817481
IVS II-74 (T>G)	c.315 + 74T>G	rs7480526	178	0.7921 (n = 141)	0.2079 (n = 37)	149,020	0.560589	0.439411
IVS II-81 (C>T)	c.315 + 81C>T	rs7946748	178	0.9831 (n = 175)	0.0169 (n = 3)	149,168	0.893891	0.106109
IVS II-180 (T>C)	c.315 + 180T>C	rs529931134	178	1.00 (n = 178)	0.00 (n = 0)	149,304	0.999906	0.000094
IVS II-258 (G>A)	c.315 + 258G>A	rs1029410290	178	1.00 (n = 178)	0.00 (n = 0)	145,536	0.999993	0.000007
IVS II-337 (A>G)	c.315 + 337A>G	rs561258571	178	1.00 (n = 178)	0.00 (n = 0)	149,146	0.999980	0.000020
IVS II-666 (C>T)	c.316 − 185C>T	rs1609812	178	0.4157 (n = 74)	0.5843 (n = 104)	149,128	0.185056	0.814944
*233(G>C)	c.*233G>C	rs12788013	178	0.9831 (n = 175)	0.0169 (n = 3)	149,150	0.893979	0.106021

**Table 4 ijms-26-08872-t004:** The online functional predictions, including SpliceAI, CADD, Phylop, and PromoterAI, of the seven β-globin variants.

Variants	SpliceAI ^a^	CADD ^b^	PhyloP ^b^	PromoterAI ^c^
−206(C>G)	0.02	2.08 (Moderate Benign)	−0.029 (Moderate Benign)	−0.03
−198(A>G)	0.01	2.75 (Moderate Benign)	0.427 (Supporting Benign)	−0.03
−50(G>A)	0.01	11.9 (Moderate Benign)	1.74 (Supporting Benign)	−0.13
IVS II-180 (T>C)	0.01	5.22 (Moderate Benign)	0.7 (Supporting Benign)	-
IVS II-258 (G>A)	0	0.234 (Moderate Benign)	−0.341 (Moderate Benign)	-
IVS II-337 (A>G)	0	0.607 (Moderate Benign)	0.002 (Moderate Benign)	-
TTS +99 (G>C)	0.11	7.73 (Moderate Benign)	−0.836 (Moderate Benign)	-

^a^ Delta scores range from 0 to 1 and can be interpreted as the probability that the variant affects splicing at any position within a window around it (+/−500bp by default). The SpliceAI score is provided for 0.2 (high recall), 0.5 (recommended), and 0.8 (high precision) cutoffs. ^b^ CADD and PhyloP scores are based on thresholds and points to predict clinical variant classification. ^c^ PromoterAI scores range from −1 to 1, with 0 meaning no activity. Negative values represent under-expression, and positive values represent over-expression. A threshold of +/−0.1 is used for high sensitivity, and +/−0.5 for high precision.

## Data Availability

All relevant data are within the paper and its Appendix A. Further inquiries can be directed to the corresponding authors.

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
