# Peer review of "Genotype-Phenotype Correlation of Seven Known and Novel β-Globin Gene Variants"

_ijms, 2025, doi:10.3390/ijms26188872_

Round 1
Reviewer 1 Report
Comments and Suggestions for Authors
Thank you for the opportunity to review this manuscript. It is clearly written, well-structured, and presents the content in a logical and engaging manner. The flow between sections is smooth, which makes it easy to follow. I have only a few minor suggestions that I believe could further improve the clarity and overall quality of the work.
Abtract
It is clear and easy to understand .
Introduction
The introduction falls a bit short. The authors didn't describe beta thalasemia enough. For example, how common is this hemoglobinopathy in Thailand?
the clinical aspect (phenotype) of the disease itself?
Generally, there should be more information in the introduction about the disease.
Additionally, there should be more information regarding the beta-like globin locus.
Results
1- The authers mention that the four mutations ([β-198(A>G), βIVSII-180(T>C), βIVSII-337(A>G), and β*233(G>C)]) were not described before in thailand. Can you please add where exactly these mutations are found? Are they still found in Asia? Or are they found in the Mediterranean area? or both?
2- Following standard scientific nomenclature, gene symbols are presented in italics. Therefore, KLF1 should be italic.
Lines 94 to 96 regarding KFL1, while this is written correctly since the authors are referring to the protein 1st, but I suggest switching to have the DNA mutation 1st, then the protein to be consistent with the beta-globin mutation, which is also a DNA mutation rather than a protein.
3- While Table 1 is informative for the patients’ clinical parameters, the large number of missing values reduces its usefulness for some patients. These missing values should be addressed or explained. In addition, the table columns are too narrow, making it difficult to read; adjusting the layout, possibly by using landscape orientation, could improve clarity.
Sex and age should be in separate columns. Also, I notice not all patients have their age. If all your patients are adults, I suggest removing the age column and just mentioning in the footnote of the table that all patients are adults.
Regarding the KFL1 mutation in Table 1, it should indicate if these mutations are heterozygous to avoid any confusion (as mentioned in lines 94 to 96 some patients were carriers for KLF1 mutation)
4- line 99 the term (pure heterozygotic form) do you mean carrier for the mutation β-50(G>A) ? The term pure heterozygotic form is a bit confusing. I suggest adjusting this sentence.
5- Line 114 to 116. Does a significant p-value support this observation? There is a limitation when comparing the β-50(G>A)/βE genotype group (n = 4) with the β-28(G>A)/βE group (n = 26), as the small sample size in the former may limit the reliability of the statistical analysis. Nevertheless, this comparison is of interest, and it is recommended to perform the analysis while clearly acknowledging the limitation due to the small sample size.
6- From lines 98 to 129, the English language and overall flow could be improved. The frequent transitions between patients disrupt readability in this section. I suggest revising to improve clarity and maintain a smoother transition.
Discussion
1- Line 168 to 169 "seven known and novel variants in the β-globin gene, including β-206(C>G), β-198(A>G), β-50(G>A), βIVSII-180(T>C), βIVSII-258(G>A), βIVSII-337(A>G), and β*233(G>C), were identified". Kindly adjust this, start with the noval variant then the seven known (when the mutation listed you started with the noval one). This part need to be adjusted for clearity
2- Line 219, I suggest to start with the novel results in that sentence.
3- Line 208 "However, in the β-208 50(G>A)/β0 and β-50(G>A)/β+ genotypes, all patients revealed β-thalassemia trait phenotype, rather than severe homozygous β-thalassemia phenotype". This is important part of the discussion; however the authers didn't indicate which mutation are considered β0 ? . In line 172 it was mentioned that the noval mutation and five others should be considered as benign or likely benign. I suggest indicating they are β+ .
4- In Beta-thalassemia there are some cases where patients who have β+/β+ will present with severe phenotype but based on table 1; this doesn't seem to be the case in your study? correct?
However, I believe mentioning some of these cases in the discussion and indicating that none of the genotypes in your study have this phenotype will improve the discussion.
The same thing applies to cis mutation, which is also not the case in this study; but mentioning this in the discussion will improve the discussion.
While the study is very interesting, it has several limitations. I strongly recommend a critical appraisal of the finding with the inclusion of a separate section dedicated to discussing these limitations.
I like to thank the authers for their effort and I hope my suggestion helps improve the manscript quality.
Reviewer 2 Report
Comments and Suggestions for Authors
Dear Editor,
First, I sincerely thank you for the opportunity to review this well-constructed manuscript investigating "Genotype-phenotype correlation of seven known and novel β- 2
globin gene variants". This is a strong, clinically relevant study that provides valuable data for the field of thalassemia diagnostics and genetic counseling, particularly in a region with a high prevalence of hemoglobinopathies.
Despite these strengths, I recommend major revisions to enhance the manuscript’s consistency:
Major Comments:
- The authors need to justify that in the abstract 89 normal subjects were analyzed to confirm the phenotypic expression of the variants. However, the methodology and results sections lack a clear description of:
- How these 89 subjects were selected.
- The full results of this analysis (e.g., how many of each variant were found in this control cohort? What were their precise Hb A2 levels?).
- This is crucial for claims like "β-198(A>G) and β*233(G>C) variants were also identified in 1.69% of normal subjects," which currently lacks context.
- The novel variant is presented as β-206(C>G). The manuscript must state its official HGVS (Human Genome Variation Society) nomenclature (e.g., HBB: c.-156C>G) based on a defined transcript number. Furthermore, a condition for publication should be that this novel variant has been submitted to a public database like dbSNP or ClinVar, and the accession number must be provided in the manuscript.
Round 2
Reviewer 2 Report
Comments and Suggestions for Authors
The authors addressed my comments carefully